# Serosurvey of Nonhuman Primates in Costa Rica at the Human–Wildlife Interface Reveals High Exposure to Flaviviruses

**DOI:** 10.3390/insects12060554

**Published:** 2021-06-15

**Authors:** Andrea Chaves, Martha Piche-Ovares, Carlos N. Ibarra-Cerdeña, Eugenia Corrales-Aguilar, Gerardo Suzán, Andres Moreira-Soto, Gustavo A. Gutiérrez-Espeleta

**Affiliations:** 1School of Biology, University of Costa Rica, San José 11501-2060, Costa Rica; gustavo.gutierrez@ucr.ac.cr; 2Department of Ethology, Wildlife and Laboratory Animals, School of Veterinary and Zootechnics, National Autonomous University of Mexico, Ciudad Universitaria, Av. Universidad #3000, Mexico City 04510, Mexico; gerardosuz@gmail.com; 3Virology-CIET (Center of Research in Tropical Diseases), University of Costa Rica, San José 2060-1000, Costa Rica; maria.piche.ovares@una.ac.cr (M.P.-O.); eugenia.corrales@ucr.ac.cr (E.C.-A.); andres.moreira-soto@charite.de (A.M.-S.); 4Department of Human Ecology, Cinvestav Mérida, Yucatán 97310, Mexico; cibarra@cinvestav.mx; 5Institute of Virology, Charité-Universitätsmedizin Berlin, Corporate Member of Freie Universität Berlin Humboldt-Universität zu Berlin, and Berlin Institute of Health, 10098 Berlin, Germany

**Keywords:** neotropical non-human primates, serology, sylvatic cycles, flavivirus, dengue virus, Saint Louis encephalitis virus, West Nile virus

## Abstract

**Simple Summary:**

The presence of flavivirus-specific antibodies in neotropical non-human primates (NPs) (i.e., dengue virus) is well known. However, it is unclear if dengue virus or other flaviviruses could be maintained in sylvatic cycles. We detected the presence of antibodies against dengue virus (DENV-1, DENV-2), Saint Louis encephalitis virus (SLEV), West Nile virus (WNV), and several undetermined flaviviruses in NPs in Costa Rica. Our work suggests continuous exposure of NPs to several flaviviruses in Costa Rica. These findings open the question of whether bidirectional transmission between humans and non-human primates can occur due to human encroachment into NP habitats, the movement of NP into urban settings, or bridging vectors.

**Abstract:**

Arthropod-borne viruses belonging to the flavivirus genus possess an enormous relevance in public health. Neotropical non-human primates (NPs) have been proposed to be susceptible to flavivirus infections due to their arboreal and diurnal habits, their genetic similarity to humans, and their relative closeness to humans. However, the only known flavivirus in the American continent maintained by sylvatic cycles involving NPs is yellow fever virus (YFV), and NPs’ role as potential hosts of other flaviviruses is still unknown. Here, we examined flavivirus exposure in 86 serum samples including 83.7% samples from free-range and 16.3% from captive NPs living in flavivirus-endemic regions of Costa Rica. Serum samples were opportunistically collected throughout Costa Rica in 2000–2015. We used a highly specific micro-plaque reduction neutralization test (micro-PRNT) to determine the presence of antibodies against YFV, dengue virus 1–4 (DENV), Zika virus, West Nile virus (WNV), and Saint Louis encephalitis virus (SLEV). We found evidence of seropositive NPs with homotypic reactivity to SLEV 11.6% (10/86), DENV 10.5% (9/86), and WNV 2.3% (2/86). Heterotypic reactivity was determined in 3.5% (3/86) of individuals against DENV, 1.2% (1/86) against SLEV, and 1.2% (1/86) against WNV. We found that 13.9% (12/86) of NPs were positive for an undetermined flavivirus species. No antibodies against DENV-3, DENV-4, YFV, or ZIKV were found. This work provides compelling serological evidence of flavivirus exposure in Costa Rican NPs, in particular to DENV, SLEV, and WNV. The range of years of sampling and the region from where positives were detected coincide with those in which peaks of DENV in human populations were registered, suggesting bidirectional exposure due to human–wildlife contact or bridging vectors. Our work suggests the continuous exposure of wildlife populations to various flaviviruses of public health importance and underscores the necessity of further surveillance of flaviviruses at the human–wildlife interface in Central America.

## 1. Introduction

The *Flavivirus* genus (family Flaviviridae, arboviruses) possesses an enormous relevance in public health, exemplified by millions of cases, hospitalizations, and deaths every year [1]. Viruses such as yellow fever virus (YFV-hemorrhagic), dengue virus 1, 2, 3, and 4 (DENV 1–4-hemorrhagic), and Zika virus (ZIKV-neurotrophic) have triggered enormous outbreaks in Latin America. Additionally, the detection of West Nile virus (WNV-neurotrophic) and Saint Louis encephalitis virus (SLEV-neurotrophic) in Latin America has created a more complex epidemiologic scenario, complicating public health control measures [2,3] Despite the worldwide distribution of flaviviruses, countries in tropical and subtropical areas that possess vast forest reserves, high diversity of ecological regions, and significant fauna diversity (including vectors and hosts), represent the most favorable environments for arbovirus diversity and evolution [4,5]. Neotropical non-human primates (NPs) have been studied for decades as hosts of flaviviruses that affect humans, because NPs’ genetic and physiological characteristics are similar to those of humans, making NPs susceptible to flavivirus infections that can cross species boundaries through vectors [6]. Although in Latin America the role of NPs in the sylvatic cycles of zoonotic flaviviruses remains speculative, research has shown that NPs are classified as highly probable of being part of the sylvatic cycles of several *Flaviviruses*, such as DENV, ZIKV, and YFV [7,8].

The urban cycles of most Flaviviruses in the American continent are recognized as responsible for maintaining and causing outbreaks in human populations [9]. The only flavivirus recognized and maintained in sylvatic cycles is YFV, specifically in the South American region [9] (Chippaux and Chippaux 2018). ZIKV and DENV circulate in anthroponotic cycles, which has facilitated their rapid spread throughout the Americas [10], with no recognized sylvatic cycle. Nevertheless, possible maintenance through sylvatic cycles has been suggested in the Americas for NPs of different species, i.e., *Alouatta* spp., *Cebus* spp., *Ateles* spp., by serological and molecular detections [11,12,13]. However, these detections may be due to spillback, i.e., humans being the primary source of the virus and spilling the virus into wildlife [7,11,14,15]. Some flaviviruses are only sporadically identified, such as WNV and SLEV, which have been detected in NPs of the genera *Alouatta* and *Cebus* by molecular or serological techniques [11,13]. Both monkeys are natural reservoirs or amplifying hosts.

Sylvatic cycles of many arboviruses remain unknown in the Americas. According to investigations of YFV in Brazil, the amplification of infected NPs precedes and leads to outbreaks of short duration in human populations [9,10]. In other cases, similar to the YFV in Brazil, humans could become infected in a sylvatic transmission cycle if they invade the natural habitat, a phenomenon commonly linked to a change in land use (deforestation, agriculture, and urbanization) or to the invasion of natural landscapes (tourism, hunting) [16]. When these infected humans enter urban settings, infections could spread rapidly, transmitted by highly anthrophilic urban mosquitoes [17]. Therefore, the sylvatic transmission cycle would turn into an urban transmission cycle [18]. DENV 1–4 and ZIKV have become fully adapted to urban cycles and no longer require NPs, forest mosquitoes, or a sylvatic cycle for maintenance [4]. However, sylvatic cycles could favor the reemergence of these viruses in human populations where immunity has been reduced. In addition, for viruses such as WNV and SLEV, sylvatic cycles could provide selective environments where new strains may develop with greater or lesser virulence for humans [7,18].

NPs may become infected while resting in the treetops or while using the same feeding schedule as the vectors in other forest strata [7], allowing contact with a high diversity of vectors. Having genetic and physiological characteristics similar to those of humans, NPs are susceptible to flaviviruses that can cross species boundaries through vectors [6]. An increased contact between NPs and humans might alter flaviviral transmission cycles, leading to outbreaks in both human and NPs populations [16]. The most important human outbreaks of DENV and other emerging flaviviruses that are reported in Costa Rica occur in ecoregions where the NPs’ home ranges are located. These regions have climatic conditions favoring the presence of vectors, and human settlements favoring increased DENV burden [19]. We hypothesized that NPs captured in geographic zones with high human flavivirus infection rates might also show a high rate of flavivirus detection. We aimed at the detection of antibodies against YFV, DENV1–4, ZIKV, WNV, and SLEV in NPs of Costa Rica, using samples collected between 2000 and 2008 and in 2014 and 2015, periods of wide variation in the DENV epidemiology of Costa Rica.

## 2. Materials and Methods

### 2.1. Sampling Protocol

An observational study was conducted, in which NPs were opportunistically captured throughout Costa Rica in order to collect blood samples, between 2000 and 2008 and in 2014 and 2015 (Figure 1, Appendix A). In order to avoid collecting multiple samples from the same individual, sample identifications were verified as part of the project “Genetic, ethological and habitat studies in Costa Rica’s monkeys” from four species of native NPs: howler monkey (*Alouatta palliata*), white-face monkey (*Cebus imitator*), spider monkey (*Ateles geoffroyi*), and squirrel monkey (*Saimiri oerstedii*) (Resolution 27- 2013-National System of Conservation Areas).

This study was conducted using protocols established by the Institutional Committee for the Care and Use of Animals (Comité Institucional para el Cuidado y Uso de los Animales) of the University of Costa Rica and adheres to the legal requirements of Costa Rica and the American Society of Primatologists (ASP) Principles for the Ethical Treatment of NPs (Collection permit number: MINAET-SINAC-Costa Rica: 042–2012-SINAC). The NPs were captured using chemical immobilization darts (Type P, 1 mL, Pneu Dart Inc., William sport, PA, USA) and a compressed gas rifle (X-Caliber Gauged CO2, Pneu Dart Inc.) for individuals at long distances or with a blowgun for individuals at a close range. The anesthetics used were Zoletil 50^®^ (3.3–11 mg/kg, Virbac, Carros, France) or ketamine (5–20 mg/kg, Bremer Pharma GMbH, Warburg, Germany), in combination with xylazine (0.5–2 mg/kg) [20,21]. Following the protocol described by Glander et al. [22], the shoot distance was between 5 m and 25 m. Once the tranquilizer took effect, the animals were caught using 2 m × 2 m elastic nets placed directly below each individual animal and held by at least three people. As soon as the animal was anesthetized, a 2–4 mL blood sample was taken from the femoral, saphenous, or cephalic vein and maintained at 4 °C until arrival at the laboratory. The NPs were monitored until recovered from anesthesia and safely released at the capture site. Once in the laboratory, the samples were centrifuged at 2500 RPM for 5 min to separate the serum, then transferred to sterile 1.5 mL tubes, and stored at −20 °C until processed.

### 2.2. Serological Screening

Serum samples were tested for specific flavivirus-neutralizing antibodies by a micro-plaque reduction neutralization test (micro-PRNT) as previously described [23]. The Centers for Disease Control and Prevention (CDC) donated chimeric reference strain viruses (strains ZIKV-ATCC VR-748 and YFV 17D/SLEV CorAn 9124) for use in the micro-PRNT assays for ZIKV, YFV, DENV 1–4, WNV, and SLEV. The chimeric viruses use a YFV backbone (YFV-17D) and replace the pre-membrane (pr-M) and envelope (E) genes of YFV with prM and E of WNV, DENV 1–4, and SLEV (WNV, Flamingo 383-99DENV-1, PUO 359, DENV-2 218, DENV-3, PaH881/88, DENV-4 1228, SLEV CorAn 9124). These viruses express membrane and envelope proteins of the target viruses and the seven non-structural proteins of the YFV backbone; therefore, detection of neutralizing antibodies to the surface proteins suggests neutralization of the tested virus, similar to the chimeric vaccine technology [24]. Since they are constructed on the YFV 17D vaccine strain, the chimeric viruses present a minimal health risk and can be manipulated in a BSL-2 Lab.

All samples were heat-inactivated and tested at a final dilution of 1:20 against each virus for an initial screening. All samples were heat-inactivated at 56 °C for 30 min and tested at a screening dilution of 1:20 for each virus, since flavivirus titers in NPs tend to be low, as observed by other research groups [11,12]. Briefly, the diluted samples were mixed with an equal volume of each virus to produce an estimated 20 micro-plaque-forming units (PFU)/well. The virus–serum mix was incubated 1 hour at 37 °C in a 5% CO2 atmosphere. Then, 50 µL from the mix was added to a VERO cell monolayer (ATCC^®^ CCL-81™) and incubated for an hour. The liquid supernatant was removed, and 100 µL MEM with 2% FBS and 1.5% of carboxymethyl cellulose (CMC) was added. Each 96-well plate contained a virus control (100% virus plaques) and a positive human serum control with antibodies against YFV (determined by other methods such as commercial ELISA). After 3 days of incubation, the monolayers were fixed with formaldehyde 10% for an hour and stained with a crystal violet solution 1%. The plaques were then counted, and the samples were classified as positive to a specific flavivirus when there was a reduction of at least 90% in the formation of plaques.

Samples with titers ≥20 to any flavivirus were considered positive and selected for serial two-fold dilutions that ranged from 1:40 to 1:1280. Monotypic or heterotypic patterns were differentiated according to whether the animal was positive to one or to several flaviviruses, respectively. For heterotypic patterns, the interpretation of micro-PRNT data was as follows: animals with a neutralizing antibody titer higher than four-fold were considered positive to the antibody for that virus or viruses; animals with neutralizing antibody titers against multiple viruses without a four-fold difference in titer were considered flavivirus antibody-positive with no specific virus identified and labeled as positive to undetermined flaviviruses [11].

## 3. Results

We analyzed a total of 86 NPs’ serum samples, of which 83.7% (72/86) were from wild caught individuals belonging to three species: 94.4% (68/72) howler monkeys, 4.2% (3/72) squirrel monkeys, and 1.4% (1/72) white-faced monkey. The remaining 16.3% (14/86) of the samples were from captive individuals, all spider monkeys (Table 1, Appendix A).

### 3.1. Serological Screening

Of the total 86 NPs evaluated, 40.7% (35/86) had evidence of prior flavivirus infection, irrespective of the flavivirus detected or the presence of a monotypic or heterotypic reaction, which suggested high exposure to flaviviruses. The remaining 59.3% (51/86) of the evaluated NPs were determined to be seronegative to all flaviviruses evaluated. Based on the 95% confidence intervals of the infection status (positive or negative), no differences between sex, origin, or age were detected. The sex of two individuals and the age of 11 individuals were undetermined; all these individuals were negative to flavivirus (Table 1, Appendix A). 

No antibodies against DENV-3, DENV-4, YFV, or ZIKV were detected. A total of 24.4% (21/86) of the NPs analyzed presented homotypic reactivity when tested for the other flaviviruses: 47.6% (10/21) for SLEV (nine howler monkeys and one spider monkey), 14.3% (3/21) for DENV-1 (one howler monkey, one spider monkey, and one squirrel monkey), 28.6% (6/21) for DENV-2 (five howler monkeys, and one spider monkey), and 9.5% (2/21) for WNV (all howler monkeys) (Table 2, Figure 1A–C). A total of 19.8% (17/86) of the NPs showed heterotypic reactivity: 11.8% (2/17) of individuals for DENV-1 (all howler monkeys), 5.9% (1/17) of individuals for DENV-2 (howler monkey), 5.9% (1/17) for SLEV (howler monkey), and 5.9% (1/17) of individuals for WNV (howler monkey) (Table 2, Figure 1A,B). We found that 70.5% (12/17) of the NPs were positive for an undetermined flavivirus, as shown by the titers of neutralizing antibodies against multiple viruses without fourfold difference (10 howler monkeys, and 2 spider monkeys) (Table 2, Figure 1D).

### 3.2. Geographical and Eco-Regional Distribution of Seropositives

All samples came from four of the seven ecoregions of Costa Rica (Figure 2): Central American dry forests (CADF), with the majority of samples (*n* = 50), Isthmian-Atlantic moist forests (I-AMF; *n* = 19), Costa Rican seasonal moist forests (CRSMF; *n* = 11), and Isthmian-Pacific moist forests (IPMF; *n* = 6). Three of these ecoregions, CADF, CRSMF, and I-AMF, presented individuals with antibodies against DENV and SLEV. Still, CADF was the ecoregion where most detections occurred (DENV 77% and SLEV 63% of infections among ecoregions). The only 3 detections of WNV and the 12 positives for an undetermined flavivirus were also from this ecoregion.

## 4. Discussion

In this study, we detected neutralizing antibodies against undetermined flaviviruses, as well as putative specific antibodies against SLEV, DENV-1, DENV-2, and WNV in NPs. The detection rate is in concordance with those of other research groups analyzing antibodies against flaviviruses [11]. Our dataset suggests that both wild-caught and captive individuals are being exposed to the same flaviviruses. The NPs in our study inhabit patches of forests close to human activities, which we can classify as presenting favorable conditions for the appearance of outbreaks or for the enzootic maintenance of vector-borne viruses [10] and may represent a scenario for putative bidirectional transmission of flaviviruses. The average birth rate of howler monkeys is 0.42 per female per year, even though births are markedly seasonal, and it is common to observe births every year [25]. Therefore, wild NPs populations are often replenished with immunologically naïve individuals that could be susceptible to the flaviviruses analyzed. The protective effect and duration of maternal anti-flavivirus antibodies are poorly understood in NPs [26].

In the NPs analyzed, we detected homotypic and heterotypic reactions mainly to SLEV. The strikingly high seropositivity to SLEV through the years has important public health implications. In Costa Rica, there are no SLEV-monitoring programs, even though there is evidence of the presence of SLEV in Latin America since 1960, and human cases have been reported sporadically in Brazil and Argentina [27,28]. The detection of antibodies against SLEV in Costa Rica was reported only in another study showing a high prevalence (80%, 87/109) in two sloth species (two-toed sloth, *Choloepus hoffmanni* and three-toed sloth, *Bradypus variegatus*), using hemagglutination inhibition (HI) complement fixation (CF) tests and PRNT [29]. Since sloths could share treetop habitats with NPs, it will be important to investigate the potential vector species involved in this ecosystem. SLEV was detected in ecoregions (I-AMF, CADF, and CRSMF) with varied altitude, environmental conditions, and demographic aspects. The wide diversity of mosquitoes suggested as putative SLEV vectors (genera *Culex, Coquillettidia, Deinocerites, Mansonia, Psorophora, Sabethes*, and *Wyeomyia*) could favor a generalized distribution of SLEV in diverse ecosystems and vertical strata of the forest throughout Latin America [30]. In Latin America, the presence of antibodies against SLEV was previously determined in the *Alouatta* genus in Argentina (32%, 2001 and 1.85%, 2017 by PRNT), and Brazil (12% by the HI test) [11,31,32], corroborating our results. The high detection rate in these studies and our study in the *Alouatta* genus suggests the need for further studies involving this NP genus in the SLEV transmission cycle. Nevertheless, the high diversity of mammals (NPs, ungulates, bats, sloths, rodents, and marsupials) and birds that might be potential SLEV hosts and vectors in Latin America suggests a complex transmission cycle that could include multiple hosts and vectors [11,29,30,33].

We detected 13 individuals with DENV-neutralizing antibodies (7 DENV-1, 6 DENV-2) belonging to three species (howler, spider, and squirrel monkeys) during the years 2001 and 2014. Previously, white-faced monkeys in Costa Rica were found positive (1.4% DENV-2, 1.4% DENV-3, and 2.8% DENV-4) by polymerase chain reaction (PCR) [13]. Additionally, other research groups have found serological evidence of exposure to DENV1-4 for decades, suggesting that it is not uncommon in the neotropics [14,34]. Dengue is the most important vector-borne pathogen in Costa Rica. Due to lack of control in the proliferation of its main vector, *Ae. aegypti* [35], the incidence of at least three types of DENV (DENV-1, 2, and 3) has increased in the last 25 years. The biggest DENV outbreaks in Costa Rica were reported in 2005, 2010, and 2013 [36]. Although we were unable to collect samples during these years, the detection of putative DENV-seropositive NPs in almost every year sampled suggests constant direct or indirect contact between NP and DENV vectors or hosts. Nevertheless, we are unable to link our detected low PRNT titers with DENV transmission or to affirm the role of NP in DENV transmission, since that requires experimental transmission.

The ecoregion where most cases of dengue in humans were registered in the years of our NPs sampling was CRSMF, with a rate of 0.28–0.36% per 1000 habitants according to the Costa Rican Health Ministry [37]. Our findings highlight the need for a research project that includes NP populations assessing their involvement in DENV transmission and maintenance. Historically, the regions with the highest reported incidence of dengue fever in humans are Atlantic, Northern, and Central Pacific coast areas [38]. These areas correspond to the CADF, CRSMF, and the I-PMF NP-seropositive ecoregions, are situated near the coast, have elevated temperatures (maximum peaks of transmission between 26 and 29 °C), and present an elevated index of human poverty [19]. These regions have climatic conditions favoring the presence of vectors and human settlements which support an increased DENV burden [19]. For that reason, the circulation in NPs poses the question of whether this is one more case of spillback or evidence that NPs may have a role in the maintenance of DENV in possible sylvatic cycles. Although none of these two options has been ruled out, the presence of vector mosquitoes inhabiting neotropical forests (for example, *Haemagogus leucocelaenus*, jungle of Brazil 2002, confirmed with DENV-1 by isolation) may favor virus transmission between infected and non-infected NPs [39]. Understanding how the climate drives the presence of these flaviviruses allows us to identify the regions where the disease is likely to persist and may emerge or resurface in the future [40].

We found antibodies against WNV in two pacific coast howler monkeys. Dolz et al. [13] detected WNV by PCR (4.2%) in the same species and same geographical setting. In addition, antibodies against WNV (15%) were reported in two-toed sloths in Costa Rica [29]. Likewise, in South America (Argentina), neutralizing antibodies against WNV were determined in 22.2% of the howler monkeys analyzed [11]. Although WNV is maintained in enzootic cycles in birds and mosquitoes, a wide variety of other vertebrate species appear to be susceptible to the infection, although very few alternative hosts appear to develop viremia with sufficient load to favor the transmission of the virus [41].

None of the NPs in this study were found positive to either YFV or ZIKV. This suggests either low circulation of both agents in the samples analyzed or that the high susceptibility of several species of NPs to YFV leads to a high mortality, and therefore individuals are taken out of the population [10]. Central America has been free of YFV since the sixties as a result of the control of *Aedes* spp. for a period of time with the use of DDT and of massive vaccination. To date, there is no evidence of new cases in the area [9]. In addition, our results are in accordance with the time of introduction in 2016 of ZIKV to Costa Rica, which suggests validity of our methodology, since no unspecific reactivity was detected. ZIKV possible maintenance through sylvatic cycles has been suggested in America, with NPs being the wild host primarily involved [7]. However, it has not been conclusively demonstrated, and it is suggested that the infection is detected in NP species (for example, *Alouatta* sp.) possibly due to spillback [15].

Limitations of our study include the heterogeneity in the number of individuals sampled per year and the number of NP species sampled; however, these are common limitations in studies with free-ranging NPs due to their high mobility and difficulties with capture. Therefore, we cannot exclude a sampling bias that limits the conclusions of our work. We emphasize that in Latin American countries with few resources for research, biological sample banks provide surveys that give insights for the creation of research questions for future investigations and generate primary useful information that can be used for macroecological studies of disease ecology [42]. Another limitation is that flaviviruses often cross-react with each other; therefore, we cannot exclude that sequential infections of various flaviviruses or other non-tested or unknown flaviviruses that cocirculate might have caused the reactivity observed, which could be the case for the detections with lower screening titers (1:20) [11]. The analysis of the serological results requires careful evaluation, especially when there is co-circulation of multiple flaviviruses. PRNT is one of the most specific tests available, often used to define several more closely related flavivirus sero-complexes [11]. The high detection of positive reactivity for an undetermined flavivirus in our data speaks in favor of this scenario. Nevertheless, homologous reactivity and the titer difference for a specific flavivirus from the diverse battery of flaviviruses tested speaks in favor of reactivity to the virus in question.

Our study underscores a neglected circulation of SLEV in Costa Rica. Humans from the same sites should be further tested to address if similar infection prevalence is observed and to assess if NPs may be used as sentinels of unsurveilled circulation of flaviviruses in Latin America. Additionally, our results suggest the need to further test NPs in the region for other arboviruses that have been detected in South American NPs, such as the Mayaro virus (Alphavirus) and the Oropouche virus (Orthobunyavirus), to assess their potential emergence or circulation in Central America [43,44]. In a cycle of successful flavivirus transmission, it is essential that all three components—virus, arthropod, and vertebrate—are available in sufficient quantities, at the same time and in the same location, to establish and maintain the disease. We find it necessary to determine the presence of competent vectors whose niche is at the wildlife–human interface. Vertebrates, therefore, serve as highly efficient amplification hosts in the presence of abundant competent mosquito vectors [16]. This knowledge will serve to identify local risk factors and determine whether human or modified NPs behaviors may be affecting the transmission cycle of flaviviruses [8]. This may indicate if the Costa Rican rainforests, the NPs, and the vectors that circulate in these natural ecoregions may be suitable for further uncontrolled transmission of these infectious agents.

## Figures and Tables

**Figure 1 insects-12-00554-f001:**
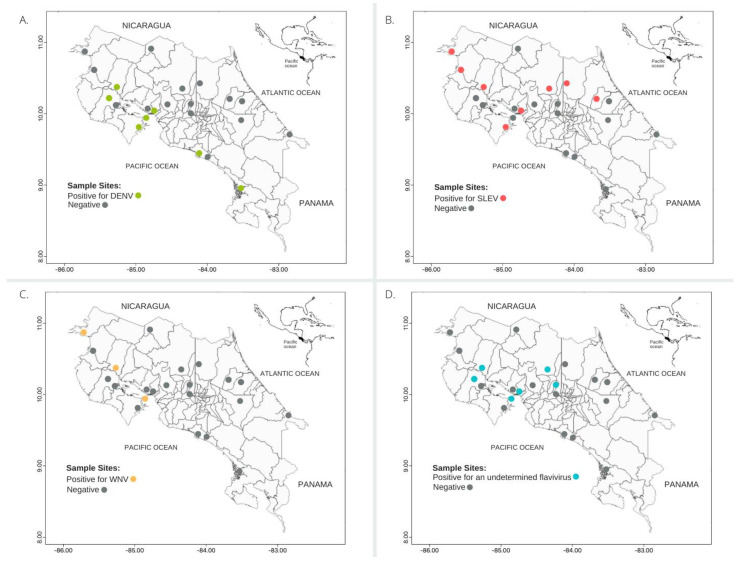
Map of (**A**) Saint Louis encephalitis virus, (**B**) dengue virus, (**C**) West Nile virus, and (**D**) undetermined flavivirus in seropositive Neotropical non-human primates from Costa Rica, 2000–2015.

**Figure 2 insects-12-00554-f002:**
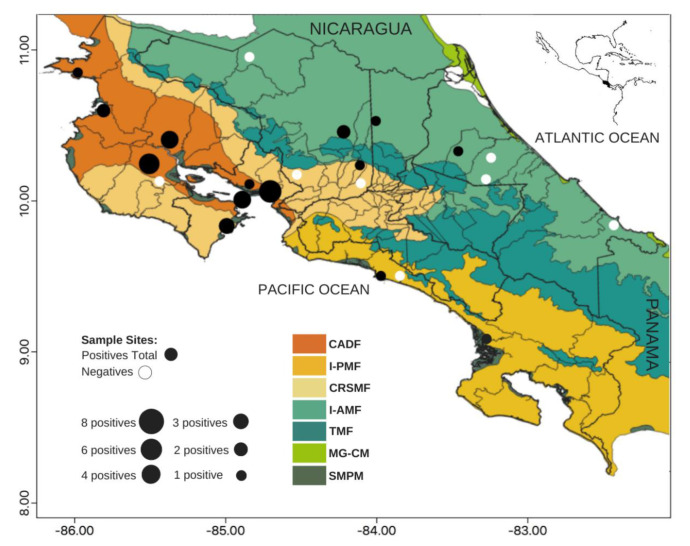
Map of Costa Rican ecoregions and flavivirus seropositive Neotropical non-human primates (NPs) from Costa Rica from 2000 to 2015. Ecoregions: CADF–Central American dry forests, CRSMF–Costa Rican seasonal moist forests, I-AMF–Isthmian-Atlantic moist forests, MG-CM—Mesoamerican Gulf-Caribbean Mangroves, TMF–Talamanca montane forests, I-PMF–Isthmian-Pacific moist forests, SMPM–Southern Mesoamerican Pacific Mangroves.

**Table 1 insects-12-00554-t001:** Number of captured Neotropical non-human primates of each species, age, sex, and origin. We show the number of individuals infected with dengue virus, Saint Louis encephalitis virus, West Nile virus, and an undetermined flavivirus. For *n* ≥ 5, infection prevalence and the binomial upper and lower 95% confidence limits are shown.

Specie	*n*	DENV	SLEV	WNV	FLAVIRUS *
*Alouatta palliata*	68	8 (0.12, 0.06–0.22)	10 (0.15, 0.08–0.25)	3 (0.04, 0.02–0.12)	10 (0.15, 0.08–0.25)
*Ateles geoffroyi*	14	2 (0.14, 0.04–0.4)	1 (0.07, 0.01–0.31)	0 (0, 0–0.22)	3 (0.21, 0.08–0.48)
*Saimiri oerstedii*	3	1	0	0	0
*Cebus imitator*	1	0	0	0	0
Age					
adult	62	11 (0.18, 0.10–0.29)	10 (0.16, 0.09–0.27)	3 (0.05, 0.02–0.13)	11 (0.18, 0.10–0.29)
juvenile	13	0 (0, 0–0.23)	1 (0.08, 0.01–0.33)	0 (0, 0–0.23)	2 (15, 0.04–0.42)
not determined	11	0 (0, 0–0.26)	0 (0, 0–0.26)	0 (0, 0–0.26)	0 (0, 0–0.26)
Sex					
female	46	3 (0.07, 0.02–0.18)	7 (0.15, 0.08–0.28)	0 (0, 0–0.08)	5 (0.11, 0.05–0.23)
male	38	8 (0.21, 0.11–0.36)	4 (0.11, 0.04–0.24)	3 (0.08, 0.03–0.21)	8 (0.21, 0.11–0.36)
not determined	2	0	0	0	0

* Positive for an undetermined flavivirus.

**Table 2 insects-12-00554-t002:** Distribution of micro-PRNT titers for flaviviruses in Neotropical non-human primates with immune pattern, Costa Rica, 2000–2015.

Animal Identification	Ecoregion	Species	Year	PRNT Titer					
				DENV-1	DENV-2	DENV-3	DENV-4	SLEV	WNV
AP-03	I-AMF	*A. palliata*	2000	Neg	Neg	Neg	Neg	1:160 *	Neg
AP-05	CADF	*A. palliata*	2000	Neg	Neg	Neg	Neg	1:160 *	Neg
AP-06	CADF	*A. palliata*	2001	Neg	Neg	Neg	Neg	1:80 *	Neg
AP-07	CADF	*A. palliata*	2001	Neg	Neg	Neg	Neg	1:40 *	Neg
AP-08	CADF	*A. palliata*	2001	Neg	Neg	Neg	Neg	1:40 *	Neg
AP-30	CADF	*A. palliata*	2001	Neg	Neg	Neg	Neg	1:40 *	Neg
AP-22	CADF	*A. palliata*	2001	1:80 ^†^	1:20 ^†^	Neg	Neg	Neg	Neg
AP-12	CADF	*A. palliata*	2001	Neg	1:20 *	Neg	Neg	Neg	Neg
AP-19	CADF	*A. palliata*	2001	Neg	1:20 *	Neg	Neg	Neg	Neg
AP-27	CADF	*A. palliata*	2001	Neg	Neg	Neg	Neg	Neg	1:20 *
AP-09	CADF	*A. palliata*	2001	1:40 ^†^	1:20 ^†^	Neg	Neg	Neg	Neg
AP-10	CADF	*A. palliata*	2001	1:20 ^†^	1:40 ^†^	Neg	Neg	Neg	1:20 ^†^
AP-11	CADF	*A. palliata*	2001	1:40 ^†^	1:40 ^†^	Neg	Neg	Neg	Neg
AP-58	SMPM	*A. palliata*	2002	1:20 *	Neg	Neg	Neg	Neg	Neg
AP-101	CADF	*A. palliata*	2003	Neg	Neg	Neg	Neg	1:160 *	1:20 *
AP-144	CADF	*A. palliata*	2006	1:40 ^†^	1:20 ^†^	1:40 ^†^	Neg	Neg	Neg
AP-142	CADF	*A. palliata*	2006	Neg	Neg	Neg	Neg	Neg	1:20 *
AP-147	I-AMF	*A. palliata*	2006	Neg	Neg	Neg	Neg	1:40 *	Neg
AG-50	CRSMF	*At. geoffroyi*	2006	1:20 ^†^	Neg	Neg	Neg	1:80 ^†^	Neg
AG-45	CRSMF	*At. geoffroyi*	2006	1:20 *	Neg	Neg	Neg	Neg	Neg
AG-46	CRSMF	*At. geoffroyi*	2006	Neg	Neg	Neg	Neg	1:20 *	Neg
SM-1	I-PMF	*S. oerstedii*	2006	1:20 *	Neg	Neg	Neg	Neg	Neg
AG-55	I-AMF	*At. geoffroyi*	2006	Neg	1:20 ^†^	Neg	Neg	1:20 ^†^	Neg
AG-55-1	I-AMF	*At. geoffroyi*	2006	Neg	Neg	Neg	Neg	1:80 ^†^	1:20 ^†^
FP-101	CADF	*A. palliata*	2014	1:640 *	1:20 *	Neg	Neg	Neg	Neg
FP-107	CADF	*A. palliata*	2014	>1:1280 *	1:20 ^†^	Neg	Neg	Neg	1:40 ^†^
FP-108	CADF	*A. palliata*	2014	1:80 ^†^	1:20 ^†^	Neg	Neg	Neg	1:20 ^†^
FPX-25	CADF	*A. palliata*	2014	Neg	1:20 *	Neg	Neg	Neg	Neg
FPX-21	CADF	*A. palliata*	2014	Neg	1:20 *	Neg	Neg	Neg	Neg
FP-109	CADF	*A. palliata*	2014	Neg	1:20 *	Neg	Neg	Neg	Neg
FP-102	CADF	*A. palliata*	2014	1:40 ^†^	1:20 ^†^	Neg	Neg	Neg	Neg
FP-103	CADF	*A. palliata*	2014	1:80 ^†^	1:20 ^†^	Neg	Neg	Neg	1:40 ^†^
MP-45	CADF	*A. palliata*	2015	Neg	Neg	Neg	Neg	1:80 *	Neg
MP-48	CADF	*A. palliata*	2015	Neg	Neg	Neg	Neg	1:40 *	Neg
MG-32	CRSMF	*At. geoffroyi*	2015	Neg	1:20 *	Neg	Neg	Neg	Neg

* Positive; **^†^** Positive for an undetermined flavivirus.

## Data Availability

The data used in this investigation are being supplied in their totality in the supplementary table.

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
