# Peer review of "Serosurvey of Nonhuman Primates in Costa Rica at the Human–Wildlife Interface Reveals High Exposure to Flaviviruses"

_insects, 2021, doi:10.3390/insects12060554_

Round 1

Reviewer 1 Report

Below is a list of my comments for consideration:

Line 149: Add "for"  after the ) and before use

Lines 173-180: If an individual has a 1:40 (WNV) and 1:1280 (DENV-1) as in FP-107, do you consider them to be positive for both WNV and DENV-1 since there is over a 4-fold separation between the two? Please specify one way or the other as part of this paragraph. 

Line 208: Delete "homotypic" from the table description as you have homotypic and heterotypic in the same table

Table 2: I would add a column for ecoregion to this table to help visualize the results better

Line 213: Add "c." to the figure description

Line 245: Reword as "Little is known if maternal..."

Line 291: Lowercase the "i" in In

Author Response

Line 149: Add "for"  after the ) and before use.

R/Added.

Lines 173-180: If an individual has a 1:40 (WNV) and 1:1280 (DENV-1) as in FP-107, do you consider them to be positive for both WNV and DENV-1 since there is over a 4-fold separation between the two? Please specify one way or the other as part of this paragraph. 

R/Added. In heterotypic patterns, interpretation of micro-PRNT data was as follows: animals with a neutralizing antibody titer ≥four-fold were considered positive to the antibody for that virus or viruses.

Line 208: Delete "homotypic" from the table description as you have homotypic and heterotypic in the same table

R/Added.

Table 2: I would add a column for ecoregion to this table to help visualize the results better

R/Added.

Line 213: Add "c." to the figure description

R/Added.

Line 245: Reword as "Little is known if maternal..."

R/Reworded.

Line 291: Lowercase the "i" in In

R/Corrected.

Reviewer 2 Report

Revisions address all comments in previous peer review. However, reading through draft revealed more opportunities for improvements, all minor.

Line 35: Rephrase sentence in abstract: “Here, we examined flavivirus exposure in 86 serum samples including X free range and Y captive NPs”

Line 81 “detection” should be plural (“detections”)

Lines 88 and 89. Style. Try to avoid beginning sentences with “It is” These sentences are wordy and indirect. For example the first sentence could be: “Sylvatic cycles of many arboviruses remain unknown in the Americas.”

Line 101 Change “in viruses” to “for viruses”

Line 109 Change “human and NPs populations equally” to “both human and NP populations”

Line 121. Change “wich” to “during which”

Line 187. “Encephalitis” should not be capitalized (also line 351). “Weste” should be “West”. In the table, a parenthesis is missing in the first row.

Line 213 Figure legend is missing “c. West Nile virus”. Delete the first “flavivirus” for d.

Line 241. I expected to find a range here, but it appears to be a mean.

Line 245. “The protective effect and duration of maternal anti-flavivirus antibodies is poorly understood in NPs.”

Line 268 For consistency in terms, use “sloths” rather than “folivores”

Line 280. “Aegypti” should not be capitalized.

Line 304 “controlled of” not “controlled by”

Line 351. “millimeter” is misspelled.

Line 423. Page number is missing.

Author Response

Line 35: Rephrase sentence in abstract: “Here, we examined flavivirus exposure in 86 serum samples including X free range and Y captive NPs”

R/Corrected.

Line 81 “detection” should be plural (“detections”)

R/Corrected.

Lines 88 and 89. Style. Try to avoid beginning sentences with “It is” These sentences are wordy and indirect. For example the first sentence could be: “Sylvatic cycles of many arboviruses remain unknown in the Americas.”

R/Corrected. Both birds are natural reservoirs or amplifying hosts. However, NP’s role as putative WNV or SLEV amplifying hosts remains unknown (Valentine et al. 2019).

Line 101 Change “in viruses” to “for viruses”

R/Corrected.

Line 109 Change “human and NPs populations equally” to “both human and NP populations”

R/Corrected.

Line 121. Change “wich” to “during which”

R/Corrected.

Line 187. “Encephalitis” should not be capitalized (also line 351). “Weste” should be “West”. In the table, a parenthesis is missing in the first row.

R/Corrected.

Line 213 Figure legend is missing “c. West Nile virus”. Delete the first “flavivirus” for d.

R/Corrected. Figure 1. Map of a. Saint Louis encephalitis virus, b. dengue virus, c. West Nile virus, and d. undetermined flavivirus in seropositive Neotropical non-human primates from Costa Rica, 2000–2015.

Line 241. I expected to find a range here, but it appears to be a mean.

R/Corrected. The average of births in howler monkeys is 0.42 per female per year,…

Line 245. “The protective effect and duration of maternal anti-flavivirus antibodies is poorly understood in NPs.”

R/Corrected.

Line 268 For consistency in terms, use “sloths” rather than “folivores”

R/Corrected.

Line 280. “Aegypti” should not be capitalized.

R/Corrected.

Line 304 “controlled of” not “controlled by”

R/Corrected.

Line 351. “millimeter” is misspelled.

R/Corrected.

Line 423. Page number is missing.

R/Corrected.

Reviewer 3 Report

  • ‘Simple Summary’ could be simpler; for the lay person/non-scientist to understand
  • Check use of commas throughout the manuscript.

ABSTRACT – State when the study was done/samples collected.  From the abstract it is not clear why you could suggest a decade of SLEV circulation.  Provide the details (# out of 86) for WNV to be consistent. 

INTRO – Careful with the flow of your justification for the study and switching between WNV/SLEV and viruses such as dengue/YFV – i.e. several flaviviruses do not have a mammalian host role, thus the rationale for investigating NPs is inconsistent.   

Line 85-87 is rather tenuous to suggest, given the main amplification hosts of WNV/SLEV are avian, and that NHP’s are unlikely to have a role given that humans are dead-end hosts for these viruses.  

Check capitalizations, e.g. of the word flavivirus.  Also, pluralization agreement (e.g. line 81, detections).  

Line 99 – say ‘nor’ instead of ‘and’.   

METHODS

Line 121 – ‘in which’

Line 122 – tie section 3.2 better into the Methods section to show where NPs were sampled from here; or refer to map figure.

Line 132 – did this involve animals darted at ground level? (vs falling down from canopy!)

Line 159 – provide details of heat inactivation (e.g. 56’C for 30mins); also what the samples were diluted in.

Lines 162 – how big is the well / state above what type of plates you were using

Line 166/167 – I would have used a two-fold virus/positive control dilution here.

Line 169 – formalin or formaldehyde? (formalin is much weaker and sometimes not effective to fix plates).    Was 3 days incubation used for all types of flavivirus?  E.g. SLEV can take longer to plaque.

Line 170 – crystal violet ‘solution’.

Line 174 – Was the animal positive for several flaviviruses, or just that any neutralizing antibodies cross-reacted with other test viruses? 

Also, what was concluded for monotypic positivity at a low titer?   How reliable is a 1:20 neutralization result (could this arise from antibodies against a further flavivirus member that was not evaluated here)?

Line 176 – It seems as though you performed end-point titers on those sera that appeared positive at a 1:20 dilution; however, your methods above do not currently detail this. Please include explanation of this.

RESULTS

Table 1 – (in both legend and the table) check spellings & capitalization of letters

What years were the juvenile NP’s that show Ab’s against SLEV captured? 

Line 195 – include test statistic value/P>0.05.

Line 199 – suggest ‘homotypic antibody reactivity when challenged against the other flaviviruses tested’.   Use ‘;’ or ‘-‘, not a ‘,’ to then continue and state the %’s.

Line 204 – 50% of the NPs showing heterotypic reactivity

Table 2 – excepting FP-101 & FP107, the PRNT titers to dengue are low. How accurately can you conclude dengue transmission occurred? Discuss.

What dengue serotypes are known to circulate within human populations in Costa Rica? How does this align with your findings?

Were the animals marked in any way?  How did you know that individual animals were not being recaptured in later years?

It is unclear how many NP were sampled in each year but from Table 2 results, it would seem there are gaps in between 2000-2003, 2006, and 2014-15.  Were these years the only year sampled; Please explain that better in methods.

It is also unclear how many NP were tested from what region, and whether that is enough to form the basis of a seroprevalence.   Granted that NHP sampling is challenging.

Figure 1 shows the distribution well of where antibody-positive NPs occurred, although there is some overlap, and it would be interesting to see how these patterns occur temporally via other detection methods (antibodies not indicative of when exposure occurred).  

Improve the legend – include c: WNV; also, be clear on what this is showing – where NPs were sampled, and which were seropositive against each of those viruses listed.

Figure 2 – The NICARAGUA really dominates this map, if it is possible to remove.   Could you produce a legend of what the depicted colors mean?  Please make the ecoregion text names stand out more.  It is challenging confusing to know which sero-positives are being talked about, or is this all flaviviruses in general?   Make the abundance circles in the legend black, to match for positives.

Line 216 – four of the seven ecoregions in Costa Rica.

This is confusing actually; how can all the samples come from 4 ecoregions, and then you mention that positive results came from additional ecoregions??  Adjust something here.

Line 222 – state how many NPs this is (86% of unspecific flavivirus-positive)

Line 223-225 – this is more for the discussion (unless detailed as part of the investigation in the methods)

DISCUSSION

Discuss cross-reactivity of immune responses to flaviviruses better. 

Explain how the relationship of cross-serotype antibodies to dengue types could affect results (DENV-1/DENV-2)

The majority of NPs were howler monkeys – how does their ecology affect your conclusions?  You mention birth rate, but highlight the role of sampling juveniles to indicate timing of/recent transmission. All your juveniles were negative to dengue – does that mean there is no current circulation of dengue strains, or that animals perhaps died.  What is the impact of exposure to each of these different flaviviruses on the NP’s themselves?  What is known about mortality/morbidity and how that might affect your results if animals suffered from infections.

How probable is human-howler monkey interaction given their niche?  One is arboreal vs one is on the ground; implications for contact/sharing vectors.   What are the likely species of mosquito that act as bridge vector(s) between humans-NPs?  (consider that different communities of mosquitoes may exist at ground vs. canopy level – include citations to show you are aware of that)

Some flaviviruses are hemorrhagic (i.e. YFV/dengue), while others (eg. Zika) are neurotrophic; mention of this distinction wasn’t incorporated anywhere

Lines 257 – ‘mosquito’ vector species; I’d say vertical strata of the forest.

Line 266 – role in the SLEV transmission cycle.  Mention why/if SLEV is important(?)

Line 272 – 'DENV-1'

Line 274-75 – ‘at’ doesn’t seem the correct work here.

Line 277 – where?   Do all 4 serotypes of DENV exist in Costa Rica/Neotropics? (not clear)

Line 278-79 – Long sentence; check capitals/commas

Line 282 – Your results suggest limited exposure, nevertheless how would be their role be assessed.  What is the role of humans creating spillback?  This was suggested as exposure in primates in Kenya I believe.

Lines 284-287 – This sentence seems somewhat stuck on the end and, if left here, requires something further, such as a suggested assessment of how NP exposure correlates with climatic, abiotic, or anthropogenic factors.

Lines 298-299 – This is somewhat tenuous a suggestion for future work, and shows a lack of understanding of WNV transmission.

Lines 306-08 – This seems as though it could be raising an interesting point, but it is not clear what.   Is ZIKV in Costa Rica?, or did it arrive post-2015/post-study? (if so, make this more explicit for the reader).    

Discuss the minimal likelihood of ZIKV having emerged from a sylvatic NP cycle.

Lines 316-9   Good

Linear 328-330 – This would be interesting!   Chikungunya also.

In terms of future direction, there is scarce mention of mosquitoes in a paper that is primarily about circulation of mosquito-borne viruses.  Knowledge of mosquito infection rates / vector species role is critical to support many of your conclusions and warrants more than the brief line in 331-332.

Otherwise, this is an interesting use of NHP samples in the region and thank you for your contribution with this work. 

Author Response

‘Simple Summary’ could be simpler; for the lay person/non-scientist to understand

Check use of commas throughout the manuscript.

R/Corrected.

ABSTRACT – State when the study was done/samples collected.  From the abstract it is not clear why you could suggest a decade of SLEV circulation.  Provide the details (# out of 86) for WNV to be consistent. 

R/Corrected. The sentence was deleted from this new version.

INTRO – Careful with the flow of your justification for the study and switching between WNV/SLEV and viruses such as dengue/YFV – i.e. several flaviviruses do not have a mammalian host role, thus the rationale for investigating NPs is inconsistent.   

R/Corrected. Some flaviviruses are only sporadically identified such as WNV and SLEV, which have been detected in NPs of the genus Alouatta spp., and Cebus spp. by molecular or serological techniques (Morales et al. 2017; Dolz et al. 2019). Both birds are natural reservoirs or amplifying hosts.

Line 85-87 is rather tenuous to suggest, given the main amplification hosts of WNV/SLEV are avian, and that NHP’s are unlikely to have a role given that humans are dead-end hosts for these viruses.  

R/Clarified. The sentence was deleted for this version.

Check capitalizations, e.g. of the word flavivirus.  Also, pluralization agreement (e.g. line 81, detections).  

R/Checked.

Line 99 – say ‘nor’ instead of ‘and’.  

R/Corrected. 

METHODS

Line 121 – ‘in which’

R/Corrected.

Line 122 – tie section 3.2 better into the Methods section to show where NPs were sampled from here; or refer to map figure.

R/Corrected. An observational study was conducted, in which NPs were opportunistically captured and in order to collect blood samples throughout Costa Rica between 2000-2008, 2014, and 2015 (Figure 1, Table S.1).

Line 132 – did this involve animals darted at ground level? (vs falling down from canopy!)

R/Clarification added. Following the protocol described by Glander et al. (1991), the shoot distance was between 5 m to 25 m. Once the tranquilizer took effect, the animals were caught using 2 m x 2 m elastic nets placed directly below each individual animal and held by at least three people.

Line 159 – provide details of heat inactivation (e.g. 56’C for 30mins); also what the samples were diluted in.

R/Added. All samples were heat-inactivated at 56°C for 30 minutes and tested at a screening dilution of 1:20 for each virus since flavivirus titers in NPs tend to be low, as observed by other research groups (Morales et al. 2017, Moreira-Soto et al. 2018a).

Lines 162 – how big is the well / state above what type of plates you were using

R/ Added. Each 96 well plate contained a virus control (100% virus plaques) and positive human serum control with antibodies against YFV (determined by other methods such as commercial ELISA).

Line 166/167 – I would have used a two-fold virus/positive control dilution here

R/Corrected.

Line 169 – formalin or formaldehyde? (formalin is much weaker and sometimes not effective to fix plates).   

R/Corrected.

Was 3 days incubation used for all types of flavivirus?  E.g. SLEV can take longer to plaque

R/ The duration of incubation was adapted to use the micro-PRNT method due to the small amount of serum, and the readability of the plate, since three days was the ideal time to do the reading. After 4 days the readability became difficult and confusing.

Line 170 – crystal violet ‘solution’

R/Added.

Line 174 – Was the animal positive for several flaviviruses, or just that any neutralizing antibodies cross-reacted with other test viruses? 

R/Clarified. Samples that have titers ≥20 to any flavivirus were considered positive and selected for a serial two-fold dilutions that ranged from 1:40-1:1280.

Also, what was concluded for monotypic positivity at a low titer?   How reliable is a 1:20 neutralization result (could this arise from antibodies against a further flavivirus member that was not evaluated here)?

R/Clarified.

Discussion: Another limitation is that flaviviruses often cross-react with each other, therefore, we cannot exclude those sequential infections of various flaviviruses, or other non-tested or unknown flaviviruses that cocirculate might have caused the reactivity observed, which could be the case at the lower screening titer detections (1:20, Morales et al. 2017).

Line 176 – It seems as though you performed end-point titers on those sera that appeared positive at a 1:20 dilution; however, your methods above do not currently detail this. Please include explanation of this.

R/Added. Samples that have titers ≥20 to any flavivirus were considered positive and selected for a serial two-fold dilutions that ranged from 1:40-1:1280. Monotypic or heterotypic patterns were differentiated according to whether the animal was positive to one or several flaviviruses, respectively. In heterotypic patterns, interpretation of micro-PRNT data was as follows: animals with a neutralizing antibody titer ≥four-fold were considered positive to the antibody for that virus or viruses. Animals with neutralizing antibody titers against multiple viruses without four-fold difference in titer were considered flavivirus antibody positive with no specific virus identified and labeled as undetermined flavivirus (Morales et al. 2017).

RESULTS

Table 1 – (in both legend and the table) check spellings & capitalization of letters

R/Corrected.

What years were the juvenile NP’s that show Ab’s against SLEV captured? 

R/The juvenile NP that had Ab’s against SLEV was captured in 2001, see table S1.

Line 195 – include test statistic value/P>0.05.

R/Included. Based on the 95% confidence intervals of the infection status (positive or negative), no differences between sex, origin, or age were detected.

Line 199 – suggest ‘homotypic antibody reactivity when challenged against the other flaviviruses tested’.   Use ‘;’ or ‘-‘, not a ‘,’ to then continue and state the %’s.

R/Corrected.

Line 204 – 50% of the NPs showing heterotypic reactivity

R/Corrected. In 70.5% (12/17) NPs results were positive for an undetermined flavivirus, because of neutralizing antibody titers against multiple viruses without fourfold difference (ten howler monkeys, and two spider monkeys) (Table 2, Figure 1D).

Table 2 – excepting FP-101 & FP107, the PRNT titers to dengue are low. How accurately can you conclude dengue transmission occurred? Discuss.

R/Added.

Discussion: Nevertheless, we cannot link our detected low PRNT titers with DENV transmission or role of NP in DENV transmission since that requires experimental transmission experiments. 

What dengue serotypes are known to circulate within human populations in Costa Rica? How does this align with your findings?

R/Added.

Discussion: The biggest DENV outbreaks in Costa Rica were reported in 2005, 2010 and 2013 (Soto-Garita et al., 20016). Although we were unable to sample during these years, the detection of putative DENV seropositive NPs in almost every year sampled suggests constant direct or indirect contact between NP and DENV vectors or hosts.

Were the animals marked in any way?  How did you know that individual animals were not being recaptured in later years?

R/Clarification added.

Materials and Methods: In order to avoid multiple samples for the same individual sample identifications were verified as part of the project,…

It is unclear how many NP were sampled in each year but from Table 2 results, it would seem there are gaps in between 2000-2003, 2006, and 2014-15.  Were these years the only year sampled; Please explain that better in methods.

R/Clarification added.

Materials and Methods: An observational study was conducted, in which NPs were opportunistically captured and in order to collect blood samples throughout Costa Rica between 2000-2008, 2014, and 2015 (Figure 1, Table S1).

It is also unclear how many NP were tested from what region, and whether that is enough to form the basis of a seroprevalence.   Granted that NHP sampling is challenging.

R/All this information was included in the supplemental information, table S1.

Figure 1 shows the distribution well of where antibody-positive NPs occurred, although there is some overlap, and it would be interesting to see how these patterns occur temporally via other detection methods (antibodies not indicative of when exposure occurred). 

R/Discussion: Limitations of our study include heterogeneity in the number of individuals sampled per year and the number of NP species sampled, however, these are common limitations in studies with free-ranging NPs due to their high mobility and difficulties with captures. Therefore, we cannot exclude a sampling bias that limits the conclusions of our work. We emphasize that in Latin American countries with few resources for research, biological sample banks provide surveys that give insights for the creation of research questions for future investigations and generate primary useful information that can be used for macroecological studies of disease ecology (Ibarra-Cerdeña et al. 2017).

Improve the legend – include c: WNV; also, be clear on what this is showing – where NPs were sampled, and which were seropositive against each of those viruses listed.

R/Added. Figure 1. Map of a. Saint Louis encephalitis virus, b. dengue virus, c. West Nile virus, and d. undetermined flavivirus in seropositive Neotropical non-human primates from Costa Rica, 2000–2015.

Figure 2 – The NICARAGUA really dominates this map, if it is possible to remove.   Could you produce a legend of what the depicted colors mean?  Please make the ecoregion text names stand out more.  It is challenging confusing to know which sero-positives are being talked about, or is this all flaviviruses in general?   Make the abundance circles in the legend black, to match for positives.

R/Corrected.

Line 216 – four of the seven ecoregions in Costa Rica.

R/Added.

This is confusing actually; how can all the samples come from 4 ecoregions, and then you mention that positive results came from additional ecoregions??  Adjust something here.

R/Clarified. All samples came from four of the seven ecoregions in Costa Rica (Figure 2): The Central American dry forests (CADF), with the majority of samples (n=50), followed by the Isthmian-Atlantic moist forests (I-AMF; n=19), the Costa Rican seasonal moist forests (CRSMF; n=11), and the Isthmian-Pacific moist forests (IPMF; n=6). Three of these ecoregions, CADF, CRSMF, and I-AMF, presented individuals with antibodies against DENV and SLEV. Still, CADF was the ecoregion where most detections occurred (DENV 77% and SLEV 63% of infections among ecoregions). The only three detections of WNV and the 12 positives for an undetermined flavivirus were also found in this ecoregion.

Line 222 – state how many NPs this is (86% of unspecific flavivirus-positive)

R/Corrected

Line 223-225 – this is more for the discussion (unless detailed as part of the investigation in the methods)

R/Corrected. The information was included in the discussion.

DISCUSSION

Discuss cross-reactivity of immune responses to flaviviruses better. 

R/Discussed. Another limitation is that flaviviruses often cross-react with each other, therefore, we cannot exclude those sequential infections of various flaviviruses, or other non-tested or unknown flaviviruses that cocirculate might have caused the reactivity observed, which could be the case at the lower screening titer detections (1:20, Morales et al. 2017). Analysis of the serological results requires careful evaluation, especially when there is co-circulation of multiple flaviviruses. PRNT is one of the most specific tests available, often used to define several more closely related flavivirus sero-complexes (Morales et al. 2018). The high detection of positive for an undetermined flavivirus reactivity in our data speaks in favor of this scenario. Nevertheless, the homologous reactivity and the titer difference to one flavivirus from the diverse battery of flaviviruses tested speaks in favor of reactivity to the virus in question. Also, sequential infections with other flaviviruses cause strong cross-reaction anamnestic reactions, which can confer immunity, especially within members of the same serogroup, making specific serological differentiation difficult (Morales et al. 2018).

Explain how the relationship of cross-serotype antibodies to dengue types could affect results (DENV-1/DENV-2)

R/Added. Dengue is the most important vector-borne pathogen in Costa Rica. Due to lack of control in the proliferation of its main vector, Ae. aegypti (Troyo et al. 2006), dengue has had an increased incidence in the last 25 years for at least 3 types of DENV (DENV-1,2 and 3). The biggest DENV outbreaks in Costa Rica were reported in 2005, 2010 and 2013 (Soto-Garita et al., 20016). Although we were unable to sample during these years, the detection of putative DENV seropositive NPs in almost every year sampled suggests constant direct or indirect contact between NP and DENV vectors or hosts. Nevertheless, we are unable link our detected to low PRNT titers with DENV transmission or the role of NP in DENV transmission since that requires experimental transmission. 

The majority of NPs were howler monkeys – how does their ecology affect your conclusions?  You mention birth rate, but highlight the role of sampling juveniles to indicate timing of/recent transmission. All your juveniles were negative to dengue – does that mean there is no current circulation of dengue strains, or that animals perhaps died.  What is the impact of exposure to each of these different flaviviruses on the NP’s themselves?  What is known about mortality/morbidity and how that might affect your results if animals suffered from infections.

R/The impact on the health of NPs from detected exposure to Flavivirus is not known. Also, given the sample size for juvenile howler monkeys (N=10), we can estimate using the upper 95% limit for confidence interval as much as 28% of prevalence. We know that this estimation is based on relatively low sample size; however, we cannot negate the possibility of current dengue circulation in juveniles from these calculations.

How probable is human-howler monkey interaction given their niche?  One is arboreal vs one is on the ground; implications for contact/sharing vectors.   What are the likely species of mosquito that act as bridge vector(s) between humans-NPs?  (consider that different communities of mosquitoes may exist at ground vs. canopy level – include citations to show you are aware of that)

R/This information is part of the introduction.

NPs may become infected while resting in the treetops or while using the same feeding schedule as the vectors in other forest stratas (Valentine 2019), allowing contact with a high diversity of vectors. Having genetic and physiological characteristics similar to those of humans, NPs are susceptible to flaviviruses that can cross species boundaries through vectors (Wolfe et al. 1998). An increased contact between NPs and humans might alter flaviviral transmission cycles that lead to outbreaks in both human and NPs populations (Weaver & Barret 2004).  

Some flaviviruses are hemorrhagic (i.e. YFV/dengue), while others (eg. Zika) are neurotrophic; mention of this distinction wasn’t incorporated anywhere

R/ Added. Viruses such as yellow fever virus (YFV - hemorrhagic), dengue virus 1, 2, 3, and 4 (DENV 1-4 - hemorrhagic), and Zika virus (ZIKV - neurotrophic) have triggered enormous outbreaks in Latin America. Additionally, the detection of West Nile virus (WNV - neurotrophic), and Saint Louis encephalitis virus (SLEV - neurotrophic) in Latin America…

Lines 257 – ‘mosquito’ vector species; I’d say vertical strata of the forest.

R/Added. …in diverse ecosystems and vertical strata of the forest throughout Latin America (Kopp et al. 2013).

Line 266 – role in the SLEV transmission cycle.  Mention why/if SLEV is important(?)

R/This information is part of the introduction.

Introduction: In addition, for viruses such as WNV and SLEV, sylvatic cycles could provide selective environments where new strains may develop with greater or lesser virulence for humans (Lounibos 2002, Valentine 2019).

Line 272 – 'DENV-1'

R/Corrected.

Line 274-75 – ‘at’ doesn’t seem the correct work here.

R/Corrected.

Line 277 – where?   Do all 4 serotypes of DENV exist in Costa Rica/Neotropics? (not clear)

R/Clarified….dengue has had an increased incidence in the last 25 years for at least 3 types of DENV (DENV-1,2 and 3).

Line 278-79 – Long sentence; check capitals/commas

R/Corrected.

Due to lack of control in the proliferation of its main vector, Ae. aegypti (Troyo et al. 2006), dengue has had an increased incidence in the last 25 years for at least 3 types of DENV (DENV-1,2 and 3).

Line 282 – Your results suggest limited exposure, nevertheless how would be their role be assessed.  What is the role of humans creating spillback?  This was suggested as exposure in primates in Kenya I believe.

R/Added. These regions have climatic conditions favoring the presence of vectors and human settlements which support an increased DENV burden (Mena et al. 2011). For that reason, the circulation in NPs presents the question of whether they are one more case of spillback or that they may have a role in the maintenance of DENV in possible sylvatic cycles. Although the possibility has not been ruled out, the presence of vector mosquitoes inhabiting the neotropical forests (for example: Haemagogus leucocelaenus, jungle of Brazil 2002, confirmed with DENV-1 by isolation) may favor the virus transmition between infected and non-infected NPs (De Figueiredo et al. 2010). Understanding how the climate drives the presence of these flaviviruses allows us to identify the regions where the disease is likely to persist and may emerge or resurface in the future (Harris et al. 2019).

Lines 284-287 – This sentence seems somewhat stuck on the end and, if left here, requires something further, such as a suggested assessment of how NP exposure correlates with climatic, abiotic, or anthropogenic factors.

R/Added. These regions have climatic conditions favoring the presence of vectors and human settlements which support an increased DENV burden (Mena et al. 2011).

Lines 298-299 – This is somewhat tenuous a suggestion for future work, and shows a lack of understanding of WNV transmission.

R/Corrected. The information was deleted from this new version.

Lines 306-08 – This seems as though it could be raising an interesting point, but it is not clear what.   Is ZIKV in Costa Rica?, or did it arrive post-2015/post-study? (if so, make this more explicit for the reader).    

R/Clarified. Therefore, our results are in concordance with the time of introduction in 2016 of ZIKV to Costa Rica,…

Discuss the minimal likelihood of ZIKV having emerged from a sylvatic NP cycle.

R/Added. ZIKV possible maintenance through sylvatic cycles has been suggested in America, with NPs being the wild host primarily involved (Valentine et al. 2019). However, it has not been conclusively demonstrated and it is suggested that the infection is detected in NP species (for example: Alouatta sp.) possibly due to spillback (Moreira-Soto et al. 2018b).

Lines 316-9   Good

Linear 328-330 – This would be interesting!   Chikungunya also.

In terms of future direction, there is scarce mention of mosquitoes in a paper that is primarily about circulation of mosquito-borne viruses.  Knowledge of mosquito infection rates / vector species role is critical to support many of your conclusions and warrants more than the brief line in 331-332.

R/Added.  In a cycle of successful Flavivirus transmission, it is essential that all three virus, arthropod, and vertebrate components are available in sufficient quantities, at the same time and in the same location, to establish and maintain the disease. We find it necessary to determine the presence of competent vectors whose niche is in the wildlife/human interface. The presence of the vertebrates therefore serves as a highly efficient amplification host in the presence of abundant competent mosquito vectors (Weaver & Barret 2004). This knowledge will serve to identify local risk factors and determine whether human or modified NPs behavior may be affecting the transmission cycle of flaviviruses (Pandit et al. 2018). This may indicate if the Costa Rican rainforests, the NPs and the vectors that circulate in these natural ecoregions may be suitable for further uncontrolled transmission of these infectious agents.

This manuscript is a resubmission of an earlier submission. The following is a list of the peer review reports and author responses from that submission.

Round 1

Reviewer 1 Report

There was definitely some overall improvement with the manuscript, however I still feel significant improvement is needed, especially with the results section. Below are my detailed comments:

Line 23-24: "exemplified by for yellow fever virus," doesn't make sense. Please reword.

Line 25: lowercase the "E" in encephalitis

Line 48: Reword as "...where positives were detected coincide..."

Line 54: Lowercase the "E" in Encephalitis

Line 67: The first time an abbreviation is used in the main text of the body should be spelled out even if you did so previously in the abstract - "Neotropical non-human primates (NPs)..."

Line 68: Add a comma after humans

Line 72: Replace Flavivirus with Flaviviruses

Line 77: Delete "is"

Line 79, 84, 287: Unless this journal has specific guidelines, species is normally abbreviated as "spp." 

Line 81: Replace on with of

Line 90: Replace flavivirus with flaviviruses

Line 92: Abbreviate neotropical non-human primate

Line 96: Troyo et al. 2011 is cited in-text but not in the references section

Line 104: End the sentence at 2015 and start a new sentence with Sample (deleting "the")

Line 122: Insert "and stored" (...1.5mL tubes and stored at -20...)

Lines 125-128: This section is confusing. Consider reword as follows, "The Centers for Disease Control and Prevention (CDC) donated chimeric reference strain viruses (ZIKV - ATCC VR-748 and YFV 17D/SLE CorAn 9124) use in micro-PRNT assays."  It is confusing as to whether or not the DENV, WNV, SLEV strains were provided by the CDC or elsewhere. Please clarify in this section. 

Line 130: Insert commas and a space between WNV and DENV-1 strains as well as between DENV-2 and DENV-3 strains

Line 137: Reword this sentence as follows, "Initial dilutions of 1:20 were selected since flavivirus..."

Line 142: Delete both "of"

Line 143: Replace "were" with "was"

Line 146: Start sentence as, "Plates were then counted..."

Line 153, 186, 188, 190, 196: Add "non-human" between Neotropical Primates for consistency 

Table 1: Double check your results. Based on the data you have in tables 2-4, it appears that your results should be as follows - SLEV = 12 individuals positive (1 suspected positive); DENV = 4 individuals positive (14 suspected positives); WNV = 1 individual positive (1 suspected positive); and Undetermined Flavivirus = 7 individuals positive. It looks like you were counting individuals from Table 4 as positive but if you are calling them "undetermined flaviviruses" because there wasn't a 4-fold increase then you can't call them positive for a specific flavivirus as well. The only "true" positives would be from Tables 2 and 3. All values will need to be addressed accordingly. 

Table 2: You need to mark animal MC-32 as a suspected positive (SP - DENV-2) since its titer was only at 1:20. The parentheses over (SP: suspected positive) are blue - change to black.  Three of the bottom lines in the table are grey and need to be made black like the others. 

Table 3: There are 8 suspected positive samples (1:20) that are not bold and one positive (1:40) that isn't bold either. Bold these for consistency between Tables 2-4. Animal FP-107 should also have WNV listed under the "Results Interpretation" section since it is above 1:20 and more than a 4-fold difference from its DENV-1 result. The second to last line is grey and needs to be black for consistency. I also feel like the suspected positives should be listed under the RI section, again, for consistency between tables. 

Table 4: Again, for consistency, list suspected positives under the RI section. And as "undetermined flaviviruses", these should all be suspected positives since there isn't the 4-fold separation that you described in the manuscript.

Tables 2-4: Personally I feel like these tables can be combined into one table. I would also remove the column "Result Interpretation" to help the table fit a little better in the journal. Instead, I'd use an asterisk by the titer values that are classified as positive and a secondary symbol for those that are suspected positive. 

Section 3.1: This section will need to have all of its statistical analyses redone according to the new "positive" values that are determined after fixing Table 1. 

Line 163: I would add a clarifying statement, "...(35/86) had evidence of prior, or suspected prior flavivirus infection,..." since of the 35 individuals, only 23 of them were deemed positive based on your result interpretations. The other 12 were only suspected positive. 

Line 166: Based on the statistics provided, it does not appear that you compared "flavivirus positive and and negative NPs between characteristics" but rather compared characteristics between flavivirus "positive" samples. You either need to change this description or redo your statistics.

Line 167: Remove the comma after origin

Line 168: Remove the period after 17.1

Line 168: Unless this journal specifies otherwise, you typically use [ ] inside of parentheses instead of using another set of parentheses. Thus it should look like "(17.1% [6/35] positive in captive environments versus 57.1% [29/35] in wild environments;...)"

Lines 168-170: You are missing the Chi-square values. Typically when reporting Chi-square results it looks like this, x2 (degrees of freedom, N = sample size) = chi-square statistical value, p = p-value. 

Line 169: Remove "females" before "positive females" and add "positive" before "males"

Line 170: With the current data, this should be 91.4% (32/35) and 8.6% (3/35) instead of 31.4 (11/35) and 5.7% (2/35). Looks like you only used the SLEV values for this comparison. You'd need to redo your Chi-square test because I would assume this would then have a statistical difference. However, these values will most likely change after addressing which samples are actually positive from Table 1...

Line 174: Replace "higher dilutions" with "meanwhile, dilutions >1:20" as to be specific

Lines 176-180: These percentages should be out of 21 and not 86 since you are talking about how many of the homotypic reactive samples (21) were positive for each of the specific antibodies. 

Lines 181-182: Same as above. Percentages should be out of 7 and not 86. And this number should be 16, not 7 (8 positive and 8 suspected positive) - refer to comments on Table 3 above.

Line 181: Delete "suspected" before against DENV-1 since those 4 samples are considered positive based on having a titer greater than 1:20

Lines 194-196: The figure maps already have a key describing the color coding so please remove all parentheses in the figure description. 

Lines 201-203, 206: IAMF needs to be replaced with I-AMF as it is abbreviated elsewhere in the manuscript. CRMF needs to be replaced with CRSMF for the same reasons 

Line 217: I would disagree with the claim that "a high rate" of neutralizing antibodies against undetermined flaviviruses were detected. Although "high rate" is subjective, 8.1% (7/86) is not what I would consider high. 

Line 218: Delete "suspected" before WNV as one of the animals (FP-107) is considered an actual positive

Line 219: You need to show some of the data from other research groups so we can see that comparison - and cite it

Lines 226-227: What about maternal immunity?

Line 228: This first sentence is a larger font size than the rest of the manuscript

Line 228-229: Mainly from SLEV (12) as the DENV cases is actually 4

Line 232: Diaz et al. 2006 is referenced in-text but not in the references section

Line 240: Italicize the genera

Line 244: Svoboba et al. 2006 is referenced in-text but not in the references section

Line 251: Correct the number of DENV cases

Line 263: There should only be 12 positive SLEV . Also, I don't feel like you can suggest "higher exposure rate in NPs following the 2013 DENV outbreak" since you don't have the data for 2013. There could have been any percentage higher exposure in 2013 but you don't know since no samples were tested from 2013. Same for years 2008-2012 - you don't have that data to make that claim. 

Line 266: Delete the "s" on NPs

Line 269: CRMF should be CRSMF

Line 270: IPMF should be I-PMF

Line 284: Add "in this study" between "NPs" and "were"

Line 363: Year 2010 should be bolded for consistency

Lines 372-374: Fix the weird formatting and change color to black

Reviewer 2 Report

Little is known about the involvement of non-human primates in flavivirus transmission cycles in Central America, so this evaluation of exposure of Costa Rican primates to human flaviviral pathogens is welcome. The authors have made significant revisions to their initial draft and the manuscript is greatly improved. However, there are still some outstanding issues that need attention.

General Comments

  1. The writing can be streamlined, with many fewer words. Any new writing should be carefully reviewed by a native English speaker.
  2. Looking at your hypothesis statement, you seem to be testing for an association between known dengue transmission in humans and flavivirus seroprevalence in primates. This would provide evidence for spillback of dengue among primates. If you are not testing for evidence of sylvatic cycles involving primates, why focus on this issue in the introductory text?
  3. In Table 1, you lump presumptive positives with confirmed positives. This is not a good idea, given the cross reactivity among flaviviruses and the possibility of novel flaviviruses not yet discovered. Furthermore, sample sizes are small enough that breaking down results among so many subcategories is pointless. This table should only present species and confirmed positives for each virus.
  4. Focus on the significant results, which is that few (3) confirmed dengue-1 positives were detected in spite of large number of human cases, and numerous (11) confirmed SLEV antibody-positive howlers were detected.
  5. Figure 2 should show the prevalence rather than raw numbers of positives. This figure is only useful for comparing dengue activity in NPs compared to dengue activity in humans. With only 3 positive dengue samples, it probably is not necessary to have a figure. Do you have access to geographically linked den 1 human case data to evaluate whether there is any spatial coincidence between location of positive primates and positive human cases (which could be represented in this figure)?
  6. The last paragraph with conclusions should include something about the potential significance of the dengue-1 results.

Specific Comments

Line 61. What is meant by “non-systematic detection”?

Line 63-67. Awkward “flavivirus worldwide distribution”. Reorder words: “the worldwide distribution of flaviviruses”. How does this introductory phrase relate to the rest of the sentence? I would delete the sentence anyway, as it doesn’t provide relevant background to your study. Your study looks at nonhuman primate exposure to several flaviviruses that are pathogenic to humans. This sentence discusses issues of biodiversity and evolution, two topics that are not explored by this study.

Line 67-69. Not sure what you’re trying to say with this sentence. It sounds like you are referring to the use of rhesus monkeys or other primates used as animal models for studying human pathogens and their pharmaceutical interventions (antiviral drugs, vaccines, etc.). The paper you cite speculates that many emerging human pathogens may derive from non-human primates.

Line 70-71. Sentence beginning with “Although” is incomplete.

Line 71. “past research” is redundant. Delete “past”. 

 Line 72. “part of the maintenance of sylvatic cycles” is redundant. “part of sylvatic cycles” is sufficient.

Line 77. Syntax. Change to “with no recognized sylvatic cycle.”

Line 79. Extraneous words. Delete “studies through”

Line 80-81. Extraneous words. Reduce to “However, these detections may be due to spillback…”

Line 86. Specify “amplifying hosts” or “amplifiers”; should be “remains”

Line 103. Syntax. “and” is the wrong word. Replace with “in which”.

Line 104. The confirmation by molecular methods was not part of this study and need not be referenced here. Delete from this line: “the sample identifications were confirmed by molecular methods”.

Line 126, 127. Missing punctuation. Also Line 130.

Line 137. This statement does not make much sense. If you expected titers to be low, then starting at a dilution of 1:20 would cause you  to miss many samples with lower titers of 5 or 10. You had sufficient volumes of serum to screen at lower dilutions I think. This was a missed opportunity. Honestly, it might be worth taking the extra time to go back and test at dilution of 1:10, which if negative would confirm positive many of the samples with titers of 20.

Line 146. Plaques were counted, not “plates”

Line 147. Reduced to at least 90% of plaques? This should be “reduced by at least 90% of plaques”. To be clear, with a challenge of 20 pfu, any well with 0, 1 or 2 plaques would be considered positive; 3 or more plaques would be negative.

Line 149-152. You tested 72 free-ranging NPs (68 howlers, 3 squirrel monkeys and 1 white-faced monkey) and 14 captive spider monkeys.  Leave out the percentages, as these look like test results, but they are not.

Line 166-170 These are inappropriate comparisons for statistical significance. These should be odds ratios. The denominator should be the total number sampled rather than the total number of positives. For example, to look for an association between flavivirus positivity and age, you would compare 32/62 adults to 4/11 juveniles (Fisher exact test).

Line 219 “in concordance with” other studies. Specify other studies involving primates.

Line 222. Confusing sentence. Please clarify.

Line 233. Awkward wording. Change “were” to “where”

Line 242. Delete “and specifically in NPs” (redundant). This sentence only refers to Alouatta spp.

Line 263. This argument is meaningless without the denominators (the number tested prior to 2013 and after 2013). This could be a very useful comparison (comparing these two infection rates) especially if the difference is significant (use the Fisher exact test)

Line 266 Rather than a surveillance program, a research project investigating the role of howler monkeys would be more appropriate.

Line 274. I disagree with this statement. For a serology result to be “in concordance” with a PCR positive rate of 4%, I would expect the antibody rate to be 80-100% positive. Your result was 0% positive, with two suspect positive samples. It is more likely that the PCR results were erroneous (I have not evaluated that study, but lab contamination is common with PCR studies).

Line 278, It is incorrect to say that previous studies confirmed your results. If that were true, then your study would not be necessary. You can say “previous studies found similar results” or “these studies corroborate our results.” Same comment for line 244.

The Abstract will need to be updated.

Reviewer 3 Report

In the revised manuscript entitled “Flavivirus Serosurvey of Nonhuman Primates in Costa Rica” by Chaves et al, the authors present data on the presence of neutralizing antibodies against a panel of flaviviruses in sera collected from non-human neotropical primates (NP) across regions of Costa Rica from 2000-2015. The location of the NP collections is presented, and where positives were located denoted. Most of the NPs in this study were wild individuals but a group of NP from a captive preserve are also included. This study represents an important area of study looking at the potential for reservoirs of human disease within sylvatic cycles.

The extensive revisions have greatly improved the manuscript.

Reviewer 4 Report

Dear editor,

I would like to thank authors to have taken into account and repied to my comments. Their answers are suitable for me and  the new version of their article has really improved. Therefore, I would recommend publishing this article in insects.

Reviewer 5 Report

 I have no furher question or comment.